# Echo of Bayes: Learned Memory Functions Can Recover Belief States

**Juan Liévano-Karim**[*]
UC Berkeley

**Peter Koepernik**
University of Oxford

**George Konidaris**
Brown University

**Cameron Allen**
UC Berkeley

## Abstract

A classical approach to solving partially observable Markov decision processes (POMDPs) [Puterman, 2014] is maintaining a belief state—a distribution over hidden states updated iteratively using Bayesian inference. In fact, an optimal policy for a POMDP can always be expressed in terms of the belief state [Russell and Norvig, 2010]. However, explicitly computing exact posteriors is intractable beyond small-scale problems, and requires knowledge of the hidden state space and transition dynamics, which are usually unavailable to the agent. Recent work has proposed model-free algorithms that help agents learn memory functions that are useful for solving partially observable tasks [Allen et al., 2024, Koepernik et al., 2025], but these methods provide no way of interpreting what exactly is being remembered. We hypothesize that these learned memory functions are implementing approximate Bayesian inference. To investigate this, we study two environments where ground-truth state information is available to the experimenter but not to the agent. By probing the hidden states of the trained recurrent networks, we find that in both environments we can reconstruct belief state distributions that closely match the ground-truth.

## 1 Introduction

Our objective is to build agents that make rational decisions even when observations contain incomplete information about the environment. One strategy is for the agent to explicitly represent a *belief state*—a distribution over the underlying (unobserved) state of the world. However, belief updates require an accurate model of how the environment evolves and produces observations, both of which depend on hidden state information that the agent cannot observe. Even if such a model were learned or provided to the agent, the Bayesian inference required to update the agent's beliefs after each new observation quickly becomes intractable for all but the smallest-scale environments.

A more scalable alternative is to train an agents' memory function—in practice a recurrent neural network (RNN)—in a model-free way, either through end-to-end reward maximization [Bakker, 2001, Hausknecht and Stone, 2015, Ni et al., 2022, Lee et al., 2025], or explicitly by introducing an auxiliary loss that penalizes non-Markovian state representations [Allen et al., 2024, Koepernik et al., 2025]. These methods do not assume agents know the set of underlying environment states, nor that they have an accurate environment model, which makes RNNs very effective for learning useful memory functions, but they provide no way of interpreting what exactly is being remembered.

We investigate whether RNNs implicitly learn to represent belief states and perform approximate Bayesian inference when trained via reinforcement learning (RL). We first train a recurrent policy with RL to maximize cumulative reward, then inspect the internal representations of the RNN to see if they contain sufficient information to reconstruct the belief state. We consider two partially observable environments, and in both, we see evidence of belief state information in the RNN representations.

---

[*]Correspondence to `jp.lievano10@uniandes.edu.co` and `camallen@berkeley.edu`.

## 2 Background

An (undiscounted) **partially observable Markov decision process** (POMDP) is a tuple

$$(\mathcal{S}, \mathcal{A}, \Omega, T, \Phi, R, H, p_0),$$

where $\mathcal{S}, \mathcal{A}$, and $\Omega$ are the state, action, and observation spaces, $T \colon \mathcal{S} \times \mathcal{A} \to \Delta\mathcal{S}$ is the transition function, $\Phi \colon \mathcal{S} \to \Delta\Omega$ is the observation function, $R \colon S \times A \to \mathbb{R}$ is the reward function, $H$ is the maximum time horizon, and $p_0 \in \Delta\mathcal{S}$ is the initial state distribution. In timestep $t$, the agent receives an observation $\omega_t \in \Omega$ and chooses an action $a_t \in A$, aiming to maximize the expected cumulative reward $\sum_{t=0}^{H} R(s_t, a_t)$. An important concept when thinking about POMDPs is the *belief state*.

The *belief state* $b_t$ is the probability distribution over underlying hidden state $s_t$, conditional on the trajectory $a_{0:t-1} = (a_0, \ldots, a_{t-1})$ of taken actions and the trajectory $\omega_{0:t} = (\omega_0, \ldots, \omega_t)$ of received observations. An important property of the belief state is that an optimal policy can always be expressed in terms of it [Russell and Norvig, 2010], in the sense that the best policy that conditions on belief state $b_t$ is as good as the best policy that conditions on entire histories $(a_{0:t-1}, \omega_{0:t})$.

Thus, an approach to solving POMDPs is to calculate a belief state, classically done through sequential Bayesian updates:

$$b_{t+1}(s|\omega_{0:t+1}, a_{0:t}) \propto \Phi(\omega_{t+1} \mid s) \sum_{s' \in \mathcal{S}} b_t(s' \mid \omega_{0:t}, a_{0:t-1}) T(s \mid s', a_t),$$

with $b_0(s) = P[s_0 = s \mid \omega_0]$. However, this approach has two crucial limitations: it requires knowledge of the true underlying dynamics $\mathcal{S}, T, \Phi$, which are usually unavailable to the agent, and the Bayesian updates are computationally intractable beyond small scale problems. We are interested in whether RNNs trained via reinforcement learning can overcome these challenges.

We hypothesize that the learned memory function of an RNN might implement approximate, model-free Bayesian inference. We investigate this by empirically probing the memory—that is, the RNN hidden state—of trained agents in two different environments. As the experimenters, we have access to the underlying dynamics and can calculate a ground truth belief state by performing exact Bayesian inference. We find that the ground truth belief state can be reliably predicted from the RNN hidden states, indicating that the RNN indeed learns to approximately encode and maintain belief states.

## 3 Environments

We briefly explain the two environments used in our experiments: CompassWorld and Marquee.

**CompassWorld**  This is a partially observable $6 \times 6$ gridworld. The agent's (hidden) *state* consists of its position $\text{pos} \in \{1, \ldots, 6\}^2$ and orientation $\text{dir} \in \{\text{N}, \text{E}, \text{S}, \text{W}\}$, and is initialized uniformly at random. The agent's *observation* specifies whether it faces one of the four walls, the goal, or empty space: $\omega_t \in \{\text{EMPTY}, \text{NORTHWALL}, \text{EASTWALL}, \text{SOUTHWALL}, \text{WESTWALL}, \text{GOAL}\}$, where the goal is placed deterministically near the center of the west wall, see Figure 1 (Left) for an illustration. At each step, the agent selects an *action* $a_t \in \{\text{MOVEFORWARD}, \text{TURNLEFT}, \text{TURNRIGHT}\}$, where MOVEFORWARD moves one cell ahead (clipped at walls), TURNLEFT rotates $90°$ counterclockwise, and TURNRIGHT rotates $90°$ clockwise. The agent receives a *reward* of $-1$ per timestep until it reaches the goal i.e. until $\omega_t = \text{GOAL}$.

**Marquee**  In this environment, a human and a robot jointly configure a row of light bulbs (like pixels in a marquee sign) of some length $L \in \mathbb{N}$. The robot's *observation* is the current light bulb configuration $\omega_t \in \{0, 1\}^L$, while the (hidden) *state* $s_t = (\omega_t, g)$ additionally contains the *goal configuration* $g \in \{0, 1\}^L$. The goal configuration is sampled uniformly at random, at the beginning of the episode, from a fixed set of possible goal states $G \subset \{0, 1\}^L$, and does not change throughout. The robots *actions* are PASS, which does nothing, and FLIP$_i$ for $i \in \{1, \ldots, L\}$, which toggles the $i$'th light bulb. If the bulb configuration matches the goal $g$ after the robot's action, then the episode ends, otherwise the robot receives a reward of $-1$ and the human, which can be thought of as part of the environment, toggles a randomly chosen bulb that does not match the goal configuration (i.e. "fixes" a random incorrect bulb). The robot's task is to (1) learn the set of possible goal states $G$

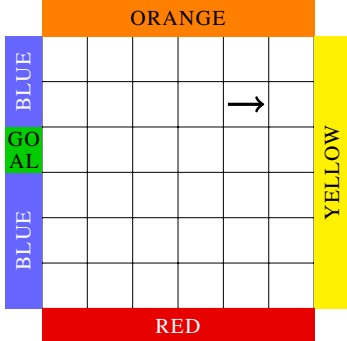

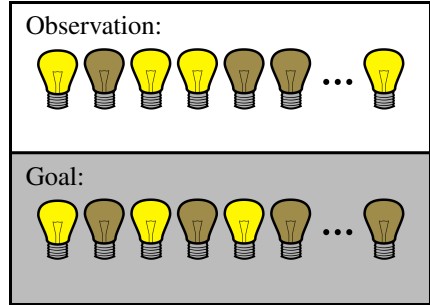

Figure 1: (Left) CompassWorld agent on position $(2, 5)$ looking East. Goal is to reach $(3, 1)$ looking West. The agent's current observation is EMPTY. (Right) Marquee domain. Top shows the current state of the light bulb array, in which light bulbs $1, 3, 4$, and $N$ are ON; bottom shows the hidden goal.

*across* episodes, and (2) infer the actual goal $g \in G$ from the human's actions *within* an episode.[2] In our experiment, the marquee sign has $L = 40$ light bulbs and the goal set has $|G| = 16$ elements, corresponding to the 16 strings obtained by repeating each 4-bit pattern $0000, 0001, \ldots, 1111$ ten times, yielding 40-bit strings. See Figure 1 (Right) for an illustration.

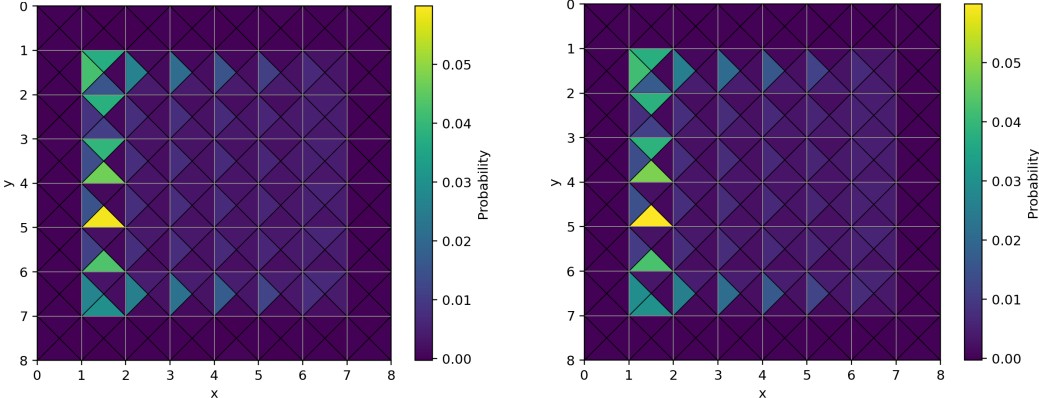

(a) Mean MLP predicted distribution over samples.   (b) Mean analytical belief distribution over samples.

Figure 2: Comparison of ground truth belief distributions and MLP-predicted belief distributions.

## 4   Experiments

**Setup**   We generate a dataset of rollouts for both CompassWorld and Marquee using recurrent policies trained with Lambda Discrepancy [Allen et al., 2024]. Each trajectory is of the form $\tau = (\omega_t, a_t, h_t, b_t)_{t=0}^{\mathrm{len}(\tau)-1}$, where $h_t \in \mathbb{R}^{\mathrm{HIDDENSIZE}}$ is the RNN hidden state produced from inputs $(\omega_t, a_{t-1}, h_{t-1})$, and $b_t \in \Delta(\mathcal{S})$ is the analytically computed belief state produced from inputs $(\omega_{0:t}, a_{0:t-1})$ (the agent does not have access to this). This yields a dataset that aligns hidden states with ground-truth beliefs. We then train an MLP $f_\theta$ to predict $b_t$ from $h_t$ using supervised learning with the KL divergence loss $L(\theta) = D_{\mathrm{KL}}(f_\theta(h_t) \| b_t)$. We include the initial pairs $(h_{-1}, b_{-1})$ with $h_{-1} = \overline{0}$ (initial memory state) and $b_{-1} = p_0$ (initial state distribution). Analogously, we also train a linear probe $g_\theta$ to predict $b_t$ from $h_t$ as above.

**Baselines**   To contextualize the performance of our probe, we compare it with two baselines: The first, 'Constant Belief' in Figures 3 and 4, always predicts a constant state distribution $\hat{b}_t \equiv p$ defined

---

[2]This environment, in which the robot infers a goal by observing human actions, is an example of an *assistance game* [Hadfield-Menell et al., 2016]. It is described here as POMDP, which is always possible for a fixed human policy [Shah et al., 2020].

for all $s \in \mathcal{S}$ as $p(s) = \text{mean}_{\tau,t} b_t^\tau(s)$ where $b_t^\tau(s)$ is the probability assigned to $s$ by the analytical belief distribution calculated from trajectory $\tau$ at time step $t$, and the mean is taken over all sampled trajectories $\tau$ and timesteps $t$. In words, $p$ is the mean of the analytical belief distributions obtained from the sampled trajectories. Importantly, this distribution also minimizes KL loss with respect to the analytical beliefs in the sense that it satisfies

$$p = \arg\min_q \frac{1}{|\mathcal{D}|} \sum_{\tau,t} D_{\text{KL}}(b_t^\tau \| q),$$

where the sum ranges over all sampled trajectories and timesteps. Therefore, this baseline can also be thought of as a linear probe trained to predict the belief state from a constant, zeroed-out input. Distribution $p$ for the CompassWorld environment is shown in Figure 2 (b).

The second baseline, 'MLP (FF)', is a feed-forward network trained to predict $b_t$ directly from the instantaneous observation $\omega_t$. Its architecture approximates the total depth of the RNN plus the hidden state linear probe: same number and width of hidden layers as the RNN (two), and an output layer of size $|\mathcal{S}|$. This baseline measures the extent to which the belief state is recoverable from the current observation alone, without any recurrence.

**Metrics**  In all of the following, $\hat{b}_t$ is the prediction of the probe, and $b_t$ is the ground truth. A *should-know timestep* is one where the ground-truth belief $b_t$ is a point mass on the true state. These are timesteps where the sequence of past observations uniquely determine the ground truth state. For example, in CompassWorld, if the agent observes EASTWALL, turns left and observes NORTHWALL, it must be in the top right corner, facing north.

1. **KL Divergence:** $D_{\text{KL}}(\hat{b}_t \| b_t) = \sum_s \hat{b}_t(s) \log_2 \frac{\hat{b}_t(s)}{b_t(s)}$, averaged across datapoints. The unit of this metric is bits, and it takes values in $[0, \infty]$, where 0 is the best.

2. **Total Variation (TV):** The $L^1$ distance, $\|\hat{b}_t - b_t\|_{L^1} = \frac{1}{2} \sum_s |\hat{b}_t(s) - b_t(s)|$, averaged across datapoints. The metric takes values in $[0, 1]$, where 0 is the best.

3. **Should-know Accuracy:** Fraction of should-know steps where the MLP predictor's argmax matches the true state. This metric does not make sense for the baseline as the uniform distribution does not have a unique argmax. Takes values in $[0, 1]$, where 1 is the best.

4. **Should-know Mass:** Probability assigned to the true state by the probe in should-know steps, averaged across data points: $\hat{b}_t(\arg\max b_t)$. Takes values in $[0, 1]$, where 1 is the best.

5. **Impossible Mass:** Total predicted probability assigned to states with zero ground-truth probability, averaged across datapoints: $\sum_{s \in S, b_t(s)=0} \hat{b}_t(s)$. Takes values in $[0, 1]$, where 0 is the best.

**Results**  Figures 3 and 4 show the performance of the probes and baselines in CompassWorld and Marquee. Our goal is to test whether the agent's learned memory function implements an approximate belief-state update. To that end, we evaluate two probes on the RNN hidden state (a linear probe and an MLP probe) and compare them to two baselines: a constant KL-optimal predictor (the empirical mean belief) and a feed-forward network MLP (FF) trained to predict $b_t$ from the instantaneous observation $\omega_t$ using an architecture matched in depth and width to the RNN + linear probe.

Across both environments, the probes on the hidden state achieve substantially lower TV and KL than either baseline and perform far better on the proposed "should-know" metrics. The probes also place far less probability mass on impossible states than the baselines. As a visualization, Figure 2 compares the average analytical belief with the average MLP-predicted belief in CompassWorld.

These findings also clarify the role of memory. The MLP (FF) baseline, which observes only $\omega_t$, performs much worse across all metrics. This indicates that the recurrent hidden state contains additional information, beyond just the most recent observation, which allows the RNN to better estimate the belief state.

The distinction between the two probes sharpens this picture. The nonlinear MLP probe shows that the hidden state contains enough information to reconstruct the full belief distribution—the information is present in the representation. The linear probe shows that this information is not stored in a highly entangled form: a single affine map from the hidden state to $\Delta(\mathcal{S})$ already recovers the

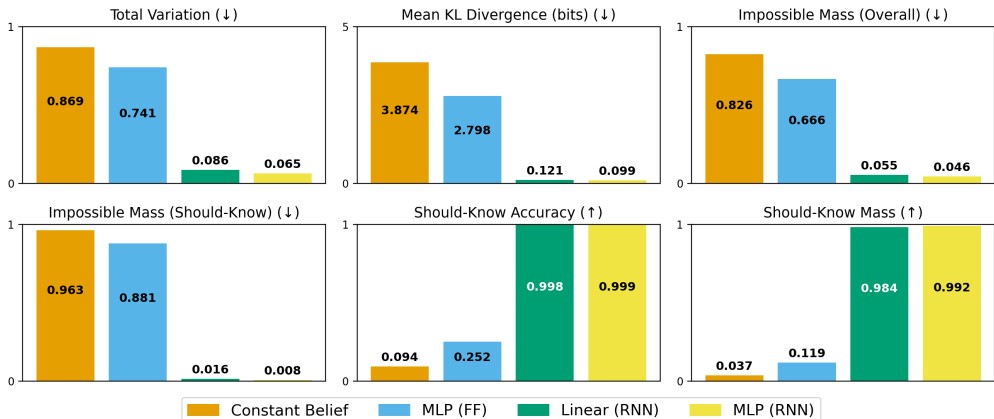

Figure 3: Belief-state estimation performance of probes and baselines in CompassWorld. Arrows indicate directionality of preference: ↑ = higher is better, ↓ = lower is better.

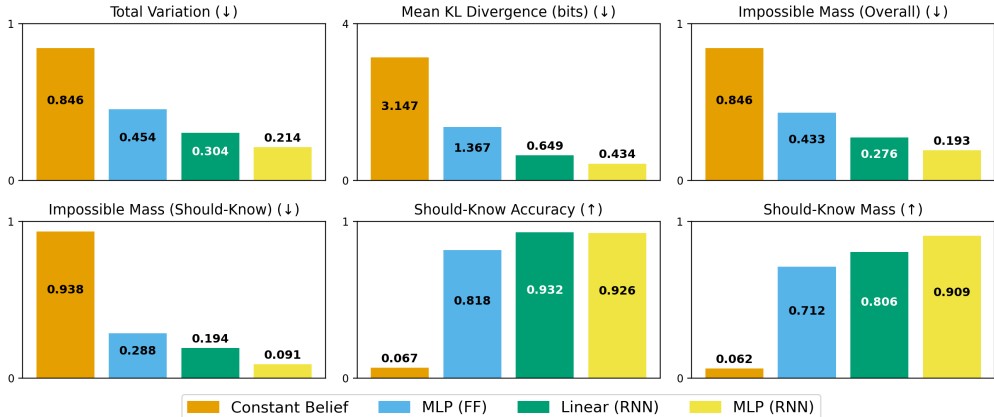

Figure 4: Belief-state estimation performance of probes and baselines in Marquee. Arrows indicate directionality of preference: ↑ = higher is better, ↓ = lower is better.

belief with high accuracy and substantially outperforms both baselines. Thus, not only does the RNN appear to maintain a belief-like summary of past observations, but this summary is linearly decodable.

Taken together, the results are consistent with our hypothesis: the trained RNN has learned an internal representation that functions like an approximate Bayesian belief state.

## 5  Future Work

We plan to extend the Marquee environment into a two dimensional grid where goals are sampled from a large library of bitmap images. This setting introduces richer spatial structure—contiguity, edges, and patterns—that should yield interesting belief state distributions. Preliminary results suggest Lambda Discrepancy [Allen et al., 2024] struggles due to the fact that long episodes entail sparse rewards. However, the Generalized Value Discrepancy (GVD) introduced by Koepernik et al. [2025] does not rely on reward signals and may succeed in such an environment. This makes the 2D Marquee environment an interesting testbed to both evaluate GVD's robustness and for extending memory probe experiments to more challenging environments.

Furthermore, in environments processed through camera inputs—such as robotic manipulation—experimenters know the hidden state (e.g., the true position of the agent or objects) but the agent only observes pixels. In such environments, the agent may create an inner representation of a belief state to manage observations that can be both partial (e.g. obscured objects) and redundant (the same state might produce different observations, e.g., because of a camera angle). It would be interesting to conduct similar experiments in these more realistic settings.

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
