# OpenReview forum: "Echo of Bayes: Learned Memory Functions Can Recover Belief States"
_NeurIPS.cc/2025/Workshop/UniReps — UniReps2025_

### Official Review · Reviewer_bUcx · 2025-09-04
**Review: RNN Memory and Bayesian Inference**

**Confidence:** 2

**Review:**

The authors investigate to what extent the memory function of an RNN is implementing approximate, model-free Bayesian inference — an interesting and relevant question to study.

---

**Pros**
- The introduction and the research question are clearly stated.
- The choice of simple environments for the experiments appears reasonable to allow for calculating the belief states.

---

**Cons**
- Using the initial state distribution as a benchmark seems potentially weak for drawing conclusions. Why was this choice made? Why is this baseline enough to measure whether the RNN is implementing Bayesian inference? The paper would benefit significantly if these points were clearly and convincingly addressed.
- Why not include a regular MLP as a baseline architecture for the policy (which does not aggregate information over time like an RNN)? The learned features of the MLP could then serve as baseline hidden states for predicting the ground-truth belief states.
- For readability, the paper would benefit from a dedicated Conclusion section. To make space, the description of the environments could be shortened.

---

**Overall Assessment**
The authors address an interesting research question. However, with the current baseline setup, it is unclear whether the experiments allow for drawing strong or meaningful conclusions regarding the stated problem.

**Score:**

2

**Topic Fit:**

2

---

### Official Review · Reviewer_T84J · 2025-09-13
**This paper investigates whether recurrent neural network (RNN)-based memory functions, trained in a model-free reinforcement learning setting, implicitly approximate Bayesian belief state updates in partially observable Markov decision processes (POMDPs). The authors study two environments — CompassWorld (a gridworld navigation task) and MarqueeSignAssistant (a cooperative goal-inference game) — where ground-truth state dynamics are available to the experimenter but hidden from the agent. By training a simple probe network on the RNN hidden states to predict belief state distributions, they show that the recovered distributions closely match the analytically computed ground-truth belief states, achieving low KL divergence and total variation. These results suggest that the learned memory functions effectively maintain internal belief states, even in the absence of explicit Bayesian updates.**

**Confidence:** 4

**Review:**

Overall Evaluation

This work provides empirical evidence that learned memory representations in recurrent RL agents can approximate Bayesian belief state updates. This is an important contribution because it bridges the gap between classical model-based POMDP solutions and modern model-free RL methods. The study is well-motivated, the methodology is sound, and the results are convincing, though somewhat limited in scope and scale.

Strengths (Pros):

1. Significance & Insight

Provides clear evidence that RNN hidden states encode belief-like distributions — an important insight for interpretability in RL.

Bridges the gap between classical Bayesian approaches and modern model-free RL, advancing theoretical understanding.

Introduces a quantitative probing methodology (KL divergence, total variation, "should-know" accuracy/mass) that could be reused by future work.

2. Clarity

Well-written and easy to follow.

Clear explanation of environments, metrics, and training procedure.

Good use of visualizations (Figure 2) to show predicted vs. ground-truth distributions.

3. Methodological Soundness

Appropriate choice of environments (small enough for exact belief computation, yet nontrivial).

Use of supervised probe with KL loss is reasonable and well-justified.

Strong empirical results: >91% accuracy on should-know states, low impossible mass, improved KL divergence vs. baseline.

Weaknesses (Cons):

1. Limited Generality

Experiments are limited to two relatively simple environments; results may not generalize to large-scale or high-dimensional POMDPs (e.g., pixel-based Atari, robotics).

No ablation studies on architecture or training procedure (e.g., effect of memory size, different RNN types).

2. Lack of Causal Verification

While the probe recovers belief-like states, it remains unclear whether the agent’s policy is actually using them as belief states or whether they merely correlate with them.

3. No Comparison to Model-Based RL

The paper does not compare performance or interpretability to methods that explicitly maintain a belief state.

Overall Impression

This is a solid and insightful empirical paper. Its main strength is interpretability: it demonstrates that model-free RL agents may implicitly be doing approximate Bayesian filtering. The work opens the door for further research on using this insight for better memory architectures and explainable RL. The scope is narrow but well executed.

**Score:**

4

**Topic Fit:**

3

---

### Official Review · Reviewer_Jou5 · 2025-09-15
**Echo of Bayes: Learned Memory Functions Can Recover Belief States**

**Confidence:** 3

**Review:**

The study effectively builds on POMDP literature, highlighting the intractability of exact belief state computation in large-scale problems. The authors’ focus on model-free algorithms for learning memory functions is promising, but their lack of interpretability is a noted drawback. Their hypothesis that these functions approximate Bayesian inference is convincingly supported by reconstructing belief state distributions that align with ground-truth in two environments. The findings enhance understanding of memory function interpretability, though testing in more diverse environments could bolster the results. Further exploration of scalability would strengthen the work’s impact.

**Score:**

3

**Topic Fit:**

2

---

### Official Review · Reviewer_HoTB · 2025-09-16
**Review of Echo of Bayes: Learned Memory Functions Can Recover Belief States**

**Confidence:** 4

**Review:**

This paper focuses on the interpretability of hidden states in RNNs trained on POMDPs. Specifically, the authors uses probes to inspect the learned memory functions in such RNNs, armed with the hypothesis that they capture approximate Bayesian belief states. This is reasonable, since exact Bayesian belief updating is known to be optimal among all policies in this setting. The authors focus on two environments where the belief states can be tractably computed and show that probes can read out these states reliably from the hidden states of trained RNNs in these environments.

**Strengths**
- Well written and easy to follow
- Hypothesis is well-reasoned and potentially insightful for RNN interpretability in these contexts

**Weaknesses**
- The authors use a non-linear probe from RNN hidden states to belief states, trained in a supervised manner. At best, this demonstrates that information about the belief states exists in the learned memory function, but this is not sufficient to argue that approximate Bayesian inference is how these networks are solving the task.

**Score:**

2

**Topic Fit:**

2